# Interface Characteristics and Mechanical Properties of 2024 Aluminum Alloy and 304 Stainless Steel Dissimilar Alloys FSLW Joint with Ni Interlayer

**Jun Liu** [1,†], **Ruixiu Guo** [2,†], **Peng Gong** [2,*], **Yumei Yue** [2,*], **Zhanxing Yu** [2] **and Yewei Zhang** [2]

1    Sinosteel Equipment & Engineering Co., Ltd., Beijing 100080, China
2    College of Aerospace Engineering, Shenyang Aerospace University, Shenyang 110136, China
*    Correspondence: gongpeng2020@163.com (P.G.); yueyumei@163.com (Y.Y.)
†    These authors contributed equally to this work.

**Abstract:** The composite structure of aluminum and steel (Al/steel) dissimilar metals has been applied in manufacturing industries for lightweight products, and friction stir lap welding (FSLW) has advantages for the welding of these two metals. To further enhance the strength of the Al/steel FSLW joint, a 0.02 mm thick nickel (Ni) foil was chosen as the interlayer, and slight plunging depth of a tool pin into the lower steel sheet was designed, which can respectively change the intermetallic compounds (IMCs) type and produce small welding heat. Choosing dissimilar 2024 aluminum alloy and 304 stainless steel materials as the research subject, the characteristics of the lap interface and the mechanical properties of the joint were mainly discussed. The results showed that the lap interface between the upper aluminum and lower steel sheets was made up of an $AlNi_3$ IMCs layer, hook structure and mechanical occlusion. The $AlNi_3$ IMCs layer with 2 μm thickness was in a reasonable range positive to the joint strength. When the rotating speed of the stirring tool increased from 800 to 1200 rpm and the welding speed of 75 mm/min was constant, the hook structure and the mechanical occlusion were both enhanced, and the tensile shear strength of the joint was decreased. A maximum joint tensile shear strength of 217 MPa was obtained at 800 rpm, and the strength value was 47.2 percent of that of the 2024 aluminum alloy base material. The Al/steel joint with shear fracture mode presented a brittle–ductile mixed fracture.

**Keywords:** aluminum and steel dissimilar metals; friction stir lap welding; Ni interlayer; interface characteristics; tensile shear strength

## 1. Introduction

With increasingly serious environmental and energy problems, the lightweight design of products has become a popular research topic in industrial manufacturing fields such as automobiles, rail transportation and ships [1]. The composite structures of different materials play an important role in lightweight design [2]. Aluminum alloy and steel (Al/Steel) composite structures combine the characteristics of the lower density of aluminum alloy and the high strength of steel, which is conducive to the light-weighting of products [3]. However, due to the large difference in the physical and chemical properties of these two materials, it is difficult to fabricate the high-quality Al/steel joint by welding, and how to avoid the formation of welding defects and reduce the generation of Al-Fe intermetallic compounds (IMCs) has been attracting more and more attention [4,5].

At present, welding techniques such as fusion welding, brazing and solid-phase welding are commonly used to join Al/steel composite structures [6–8]. However, due to the higher heat input during fusion welding, amounts of Al-Fe IMCs are generated, which greatly reduces the mechanical properties of Al/steel fusion welded joints. Relatively, the Al/steel brazed joint has less deformation as well as less Al-Fe IMCs generation. However, similar to fusion welding, the brazing is prone to defects such as inclusions and porosity.

Because of the characteristics of lower heat input and shorter high temperature residence time, solid-phase welding techniques, including friction stir welding (FSW), are suitable for fabricating Al/steel composite structures with no fusion welding defects and less Al-Fe IMCs [9,10].

In general, friction stir lap welding (FSLW) can be used to join Al/steel lap structures, while the aluminum material is always served as the upper sheet to avoid the generation heat during welding. For Al/steel FSLW, the welding parameters including process parameters and tool geometry parameters greatly affect the joint quality [11]. In particular, the plunging depth of the tool pin into the steel surface is of interest to researchers because it greatly affects the features of lap interface of the Al/steel FSLW joint. Joaquin et al. [12] used FSLW to join AA6063-T6 Al and LCG-S steel when the tool pin did not penetrate the Al plate and found that lap interface occurred during Al-Fe interatomic diffusion, achieving metallurgical bonding. Bruna et al. [13] obtained FSLW joints of AA5083 Al and GLD36 steel under a 0.2 mm plunging depth of the tool pin into the steel surface, and they stated that the mechanical interlocking and metallurgical bonding at the lap interface of the joint were obtained while the FeAl and Fe3Al IMCs were observed. Kimapong and Watanabe [14,15] investigated FSLW joints of A5083 Al and SS400 steel and stated that the highest tensile shear property of the joint was achieved when the plunging depth of the tool pin was 0.1 mm into the steel surface. The generation of Al-Fe IMCs is the prerequisite for effective bonding between Al and steel alloys, but increasing the plunging depth of the tool pin into the steel makes the Al-Fe IMCs thickness larger due to the elevated temperature, which then always leads to a decrease in the tensile shear strength of the joint. Thus, during Al/steel FSLW, the plunging depth of the tool pin into the steel surface can be regulated to avoid the formation of Al-Fe IMCs with larger thickness, which is beneficial to improve the mechanical properties of the joint.

In addition, changing the type of IMCs is another effective way to improve the welding quality of the Al/steel FSLW joint. Especially, the addition of a metal interlayer can change the interfacial atomic diffusion behavior and then the type and distribution of the IMCs [16–19]. Zheng et al. [17] added the Zn interlayer in the 6061 Al and 316 stainless steel FSLW joint and found that the formation of Al-Fe IMCs was replaced by generating a small amount of FeZn10 at the interface, while the generated Al-Zn mixed layer and Zn-steel mixed layer produced at the interface enhanced the mechanical properties of the joint. Ding et al. [16] used FSLW to join AlSi-coated 22MnB5 steel and AA5754 aluminum and found that the 1 μm thick Al7Fe2Si and Al5Fe2(Si) IMCs were formed at the interface. Nickel (Ni) has some characteristics including good plastic deformation, excellent atomic diffusion ability and infinitely solid-soluble ability in Fe. Thus, some scholars used the Ni interlayer to weld steel with other metals. Chen et al. [19] used a Ni interlayer to achieve a metallurgical bond between 5052 Al alloy and 201 stainless steel and found that the Ni1.1Al0.9/FeAl3 and NiAl3 were formed between the nickel foil and Al at the interface. Therefore, the addition of the Ni interlayer at the interface of the Al/Steel FSLW joint can improve the tensile properties of the joint due to the change of IMCs type.

From the above-mentioned discussions, it is known that for the FSLW technology of aluminum to steel, the high-strength lap joint can be fabricated under the rationally controlled Al-Fe IMCs layer thickness [20], and the addition of the Ni interlayer is beneficial to further enhance the loading capacity of the lap joint due to the decreased hardness of the IMCs layer [21]. In this study, choosing 2024 aluminum alloy and 304 stainless steel as the research object, the Al/Steel FSLW joint was fabricated under a microplunging depth of the tool pin into the steel surface and the addition of an Ni interlayer at the lap interface, and the interface characteristics, atomic diffusion behavior and mechanical properties of the joints were analyzed under different rotating speeds of the stirring tool.

## 2. Experiment Procedure

In the experiments, the upper and lower base materials (BMs) were the alclad 2024 aluminum alloy sheet and 304 stainless steel sheet, respectively. The FSW-3LM-4012 friction

stir welding machine (China FSW Center, Beijing, China) was used. Before welding, the overlapping area was cleaned with absolute ethanol, and the sheets were fixed on a specially designed fixture. As shown in Figure 1a, the dimensions of the sheets were 150 mm × 120 mm × 2 mm, the width of the overlapping area was 50 mm, and the thickness of the Ni foil as the interlayer was 0.02 mm. Wan et al. [22] studied the FSLW of 5052 aluminum alloy and 304 stainless steel under the tilt angles of 0°, 1.5° and 2.5°, and found that the tilt angle of 2.5° rather than 0° or 1.5° eliminated the welding defects including microvoids, cracks and even tunnel defects. In this study, the 2° tilting angle of the stirring tool was chosen, and this tool was rotated counterclockwise and moved along the weld centerline. The welding direction was parallel to the rolling direction of the BMs.

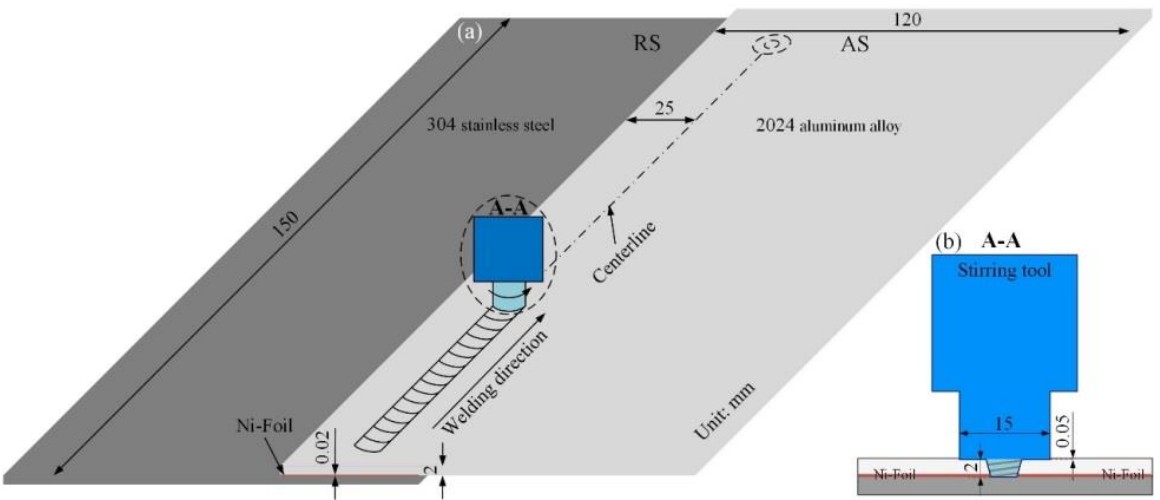

**Figure 1.** Schematic of FSLW: (**a**) welding process; (**b**) section view of A-A.

The tapered stirring tool (China Aviation Manufacturing Technology Research Institute, Beijing, China) adopted in the experiment is shown in Figure 1b, which was made of H13 steel. The shoulder diameter was 15 mm, and the pin length was 2 mm. The plunge depth of the shoulder was 0.05 mm; thus, the plunging depth of the pin tip into the steel surface was 0.03 mm. Under the condition of a welding speed of 75 mm/min, the rotating speeds were 800, 1000 and 1200 rpm, and these process parameters were designed according to our previous work and the reported results [23].

According to the standard of ISO 4136 [24], the specimens for metallographic observation and tensile shear tests were first cut by the wire-electrode cutting machine (DK7750ZC, Jiangsu Dongqing CNC Machine Tool Co., Ltd., Taizhou, China), and the cut direction was perpendicular to the welding direction. The width of the specimen for the tensile shear test was 20 mm. After polishing, the cross sections of the metallographic specimens were observed by the Olympus-GX51 optical microscope (OM) (Olympus Co., Ltd., Tokyo, Japan), and the microhardness was measured by the HVS-1000 tester (Egon Test & Measurement Instruments (Shanghai) Co., Shanghai, China) with 20 N load for a 10 s dwell time. At room temperature, the tensile shear test was executed by the 5982 universal electronic material-testing machine (Istron, Boston, MA, USA) with a loading speed of 2 mm/min. To reduce the experimental error, three tensile shear specimens were tested for each set of welding parameters, and the mean value was used to evaluate the tensile shear load of the joint. A scanning electron microscope (SEM) (HITACHI, Tokyo, Japan) was used to analyze the fracture characteristics and lap interface characteristics. The atomic diffusion and IMCs were respectively analyzed by energy dispersive spectroscopy (EDS) and X-ray diffraction (XRD) (Oxford Instruments, Abingdon, UK).

## 3. Results and Discussion

### 3.1. Cross Section Feature

The cross sections of the joints with the 0.02 mm Ni interlayer under different rotating speeds are shown in Figure 2. In this study, due to the existence of the Ni interlayer at the lap interface, the plunging depth of the pin tip into the steel upper surface is reduced from 0.05 to 0.03 mm; thus, the mechanical stirring effect on the steel interface by the stirring tool is weakened, resulting in a reduction of steel particles peeling from the initial upper steel surface. In addition, due to the good plastic deformation and excellent atomic diffusion ability [13], the Ni material can continuously migrate and fill the microgap during welding, strengthening the metallurgical bonding of the Al/steel FSLW joint. Therefore, compared with the reported results by Mahto et al. [24] and Chitturi et al. [25], there are no volume defects at the lap interface of the joint (Figure 2) under the action of the Ni interlayer, and the steel particles above the lap interface area are greatly decreased, indicating that an excellent lap interface formation of the joint is obtained.

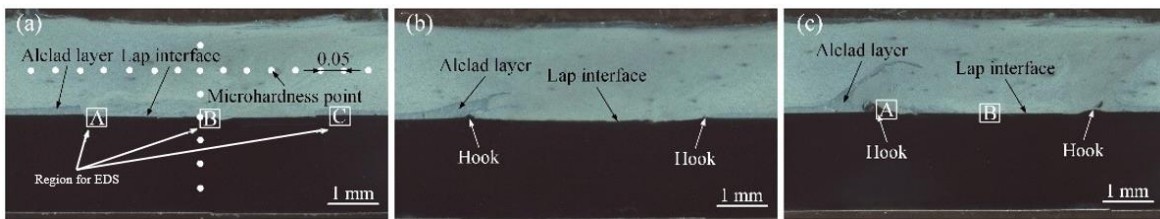

**Figure 2.** Cross sections of the joints under different rotating speeds: (**a**) 800 rpm, (**b**) 1000 rpm and (**c**) 1200 rpm.

Because the tool pin very slightly plunges into the upper surface of the steel sheet, the initial steel surface bears a stirring–extrusion effect produced by the tool pin. As a result, the upper surface of the lower steel sheet is mechanically deformed. At the same time, the upper aluminum alloy above the lap interface experiences a violent downward flow by the stirring tool, and this downward flow squeezes the lap interface. Therefore, the deformation degree of the steel surface reaches the maximum in the area on both sides of the tool pin. Due to the relatively low heat input, the upper surface of the steel at 800 rpm bears the weakest mechanical action; thus, the deformation degree of steel surface is the lowest and then the hook structure is not formed (Figure 2a). The hooks (Figure 2b,c) embedded in aluminum alloy are observed in the joint at 1000 rpm and 1200 rpm due to the increased heat input and the improved material flow behavior. These above experimental results are similar to the results reported by Liu et al. [26]. Moreover, 2024 aluminum alloy has the alclad layer on the surface, and part of alclad layer is transferred into the aluminum alloy sheet under the driving action of stirring tool, as displayed in Figure 2. With increasing the rotating speed, the distance between the highest point of alclad layer and the original lap interface is increased, because not only the material flow ability but also the driving effect of stirring tool are enhanced. Similar to the alclad layer into the upper sheet, the hook structure of FSLW joint at 1200 rpm (Figure 2c) is higher than that at 1000 rpm (Figure 2b).

### 3.2. Atomic Diffusion

The regions marked in white square in Figure 2 are selected to further observe the effect of Ni interlayer on interface microstructure of Al/steel FSLW joint. Figure 3 shows the element distribution of region A near the lap interface by EDS. During FSLW process, the tool pin constantly scrapes the upper surface of steel; thus, the steel particles are scattered in the upper aluminum alloy sheet above the lap interface, and the lap interface becomes uneven. This uneven interface is considered as the occurrence of mechanical occlusion between the upper and lower sheets. Because region A is far away from the joint center, the stirring effect of the tool pin on the Ni interlayer is weakened, resulting in the residual Ni

interlayer (Figure 3c,f). As shown in Figure 3a,b,d,e, the residual Ni interlayer effectively inhibits the diffusion of Al and Fe atoms in region A. Compared with the joint at 800 rpm, the joint at 1200 rpm bears a more intense thermo-mechanical effect by the stirring tool, which greatly reduces the amount of the residual Ni interlayer and further improves the diffusion ability of the Ni atoms due to the elevated welding temperature and the increased material flow. Therefore, for the joint at 1200 rpm, the phenomenon of Ni atoms diffused in the aluminum alloy and steel is more obvious, which is conducive to improving the metallurgical bonding of the joints.

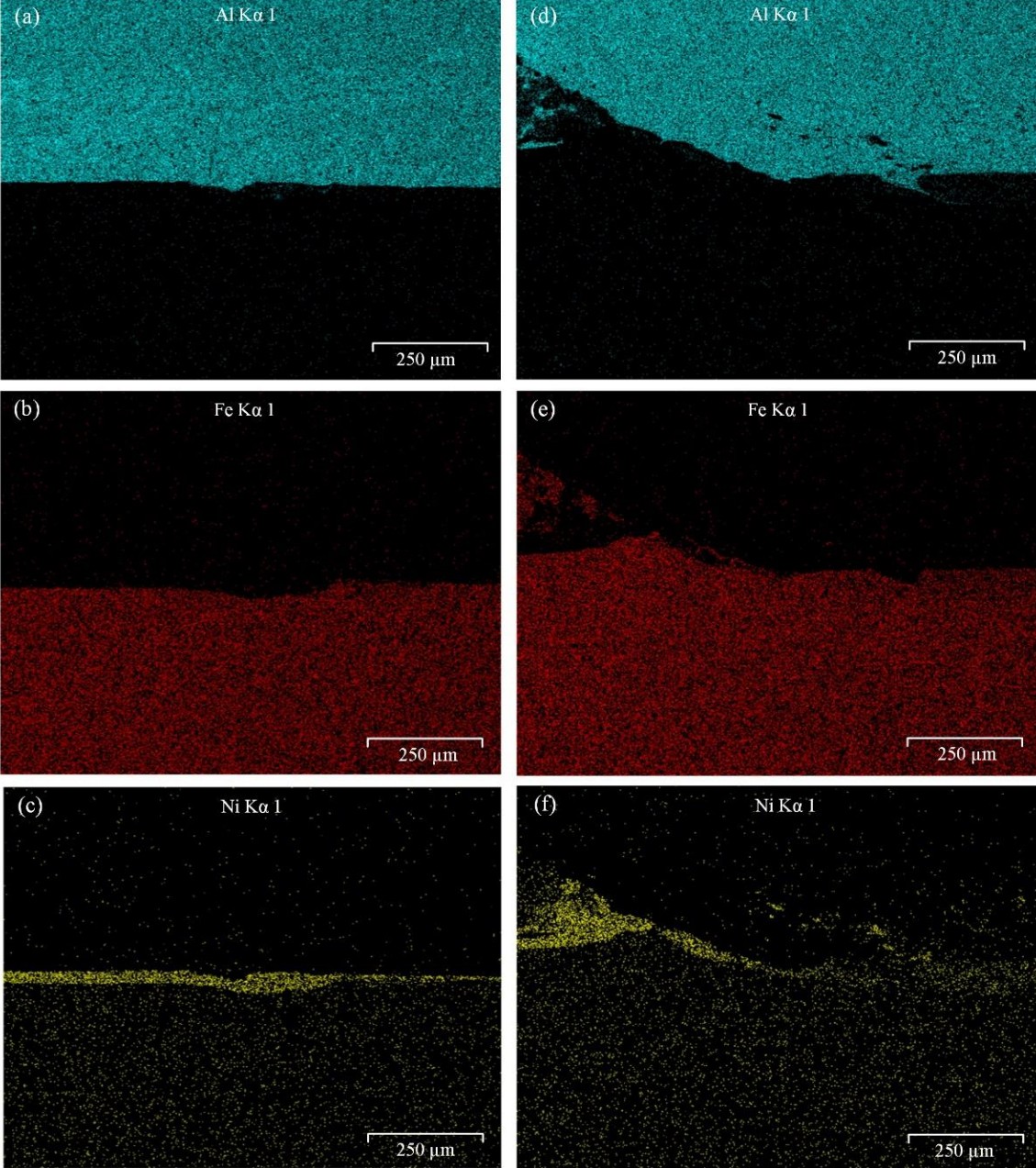

**Figure 3.** EDS analysis results of region A: (**a**–**c**) 800 rpm; (**d**–**f**) 1200 rpm.

Figure 4 shows the element distribution of region B near the lap interface by EDS. It is known that the lap interface of region B is tightly joined and the mechanical occlusion is formed at the interface. Similar to the results shown in Figure 3, the mechanical occlusion at 1200 rpm (Figure 4d,e) is enhanced compared with that at 800 rpm (Figure 4a,b), which

is closely related to the difference in the thermo-mechanical effect by the stirring tool. Compared with region A (Figure 3c,f), region B has a significantly increased diffusion degree of Ni atoms (Figure 4c,f) due to the increased welding temperature. In addition, according to the Fe-Ni phase diagram, when the temperature exceeds 345 °C, the Ni and Fe elements can form an infinite solid solution [27]. Therefore, in region B, the diffusion degree of the Ni atoms into the steel is much higher than that into the aluminum alloy.

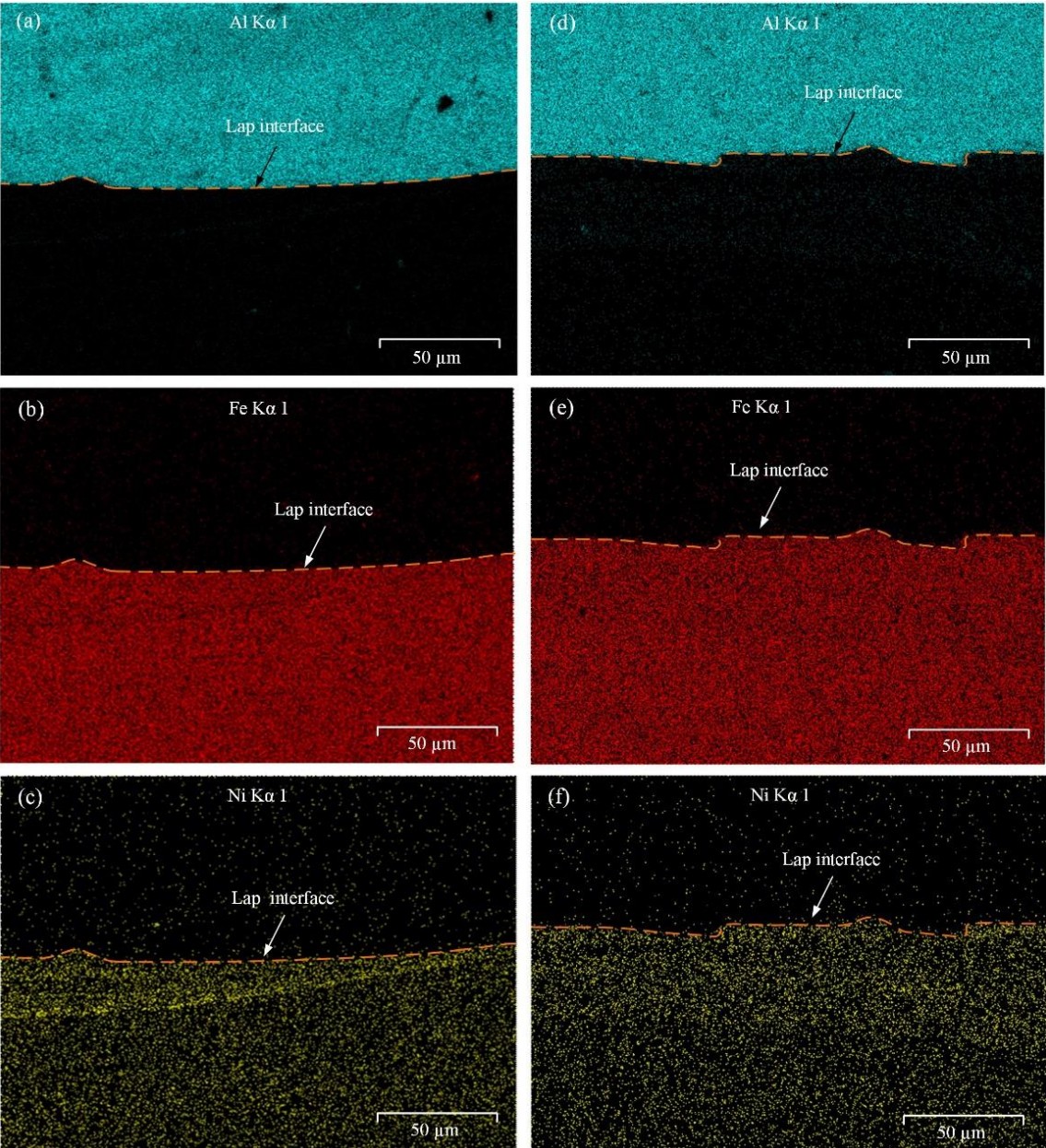

**Figure 4.** EDS analysis results of region B: (**a–c**) 800 rpm; (**d–f**) 1200 rpm.

After enlarging region B, the detailed SEM analysis in Figure 5 reveals the presence of IMCs layer at the lap interface, indicating an effective metallurgical bonding between these two metals. It is known from Figure 4 that the tool pin makes the Ni interlayer completely broken and Ni atoms fully dissolve into the steel; thus, the IMCs layers in Figure 5a,b at region B are very thin. As mentioned above, the upper surface of steel is uneven, and then, the mechanical occlusion is formed. Thus, the thickness of the IMCs layer along the lap interface is uneven, as displayed in Figure 5c,d. This uneven degree

increases under a higher rotating speed, and the maximum thicknesses at 800 and 1200 rpm are respectively 1.27 and 1.85 μm. Thus, the maximum thicknesses of the IMCs layer are increased under the higher rotating speed due to the evaluated welding temperature. The changing law of the IMCs layer thickness with the rotating speed well agrees with the results in Figure 4. Dong et al. [20] stated that there was no adverse effect on the joint quality when the thickness of the Fe-Al IMC layer was less than 2 μm. Geng et al. [28] dissimilar friction stir lap welded the 5052 Al alloy and DP590 steel and stated that the Al-Fe IMCs layer thickness beyond 1.5 μm was harmful to the joint strength. Wang et al. [29] performed FSLW of the high-strength interstitial free steel and Al–Mg–Si alloy with a 0.1 mm Ni interlayer and found that the tensile-shear load of the lap joint with a 1.8 μm thick IMCs was enhanced compared with the joint without the introduction of the Ni interlayer. For the FSLW joint of the aluminum to steel, although the IMCs layer with large thickness is harmful to the joint strength, the rational IMCs layer thickness for maximizing the joint strength is decided by many factors, including the IMCs type, the BM type, the thickness of the aluminum sheet, the effective bonding width at the lap interface and so on. In this study, the maximum thickness of the IMCs layers is less than 2 μm, which reveals that the IMCs thickness is reasonably controlled under the rational plunging depth of the tool pin into the lower steel sheet and the rational rotating speed of the stirring tool.

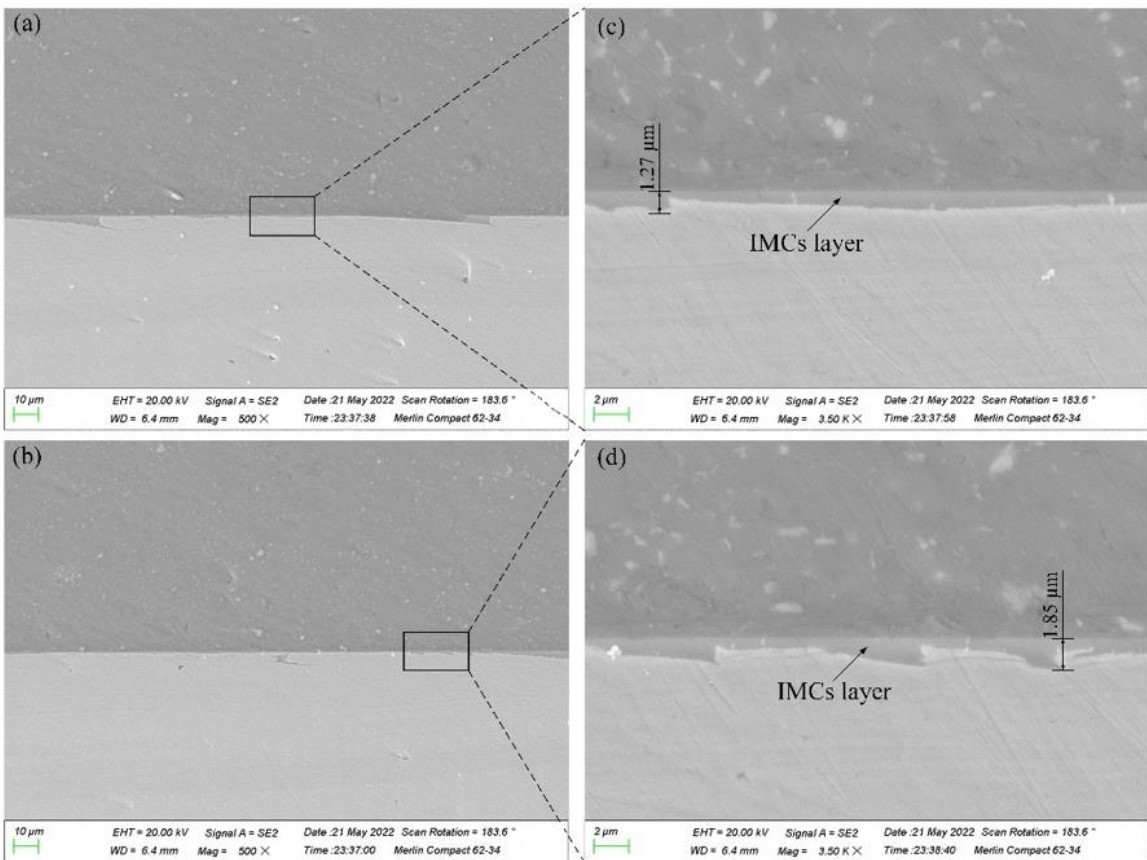

**Figure 5.** SEM analysis at region B: (**a**,**b**) 800 rpm; (**c**,**d**) 1200 rpm.

The element distribution of region C at the lap interface marked in Figure 2a is scanned, and the results are shown in Figure 6. Because region C is far away from the joint center, it does not undergo a mechanical effect of the stirring tool, the corresponding lap interface is undeformed and then straight, and the continuously distributed Ni interlayer hinders the diffusion of Al and Fe atoms (Figure 6a). Moreover, atomic diffusion on both sides of the Ni interlayer occurs, and the diffusion distance (Figure 6b) is much smaller compared with the results in Figure 4 due to the smaller temperature experienced by the materials at region C.

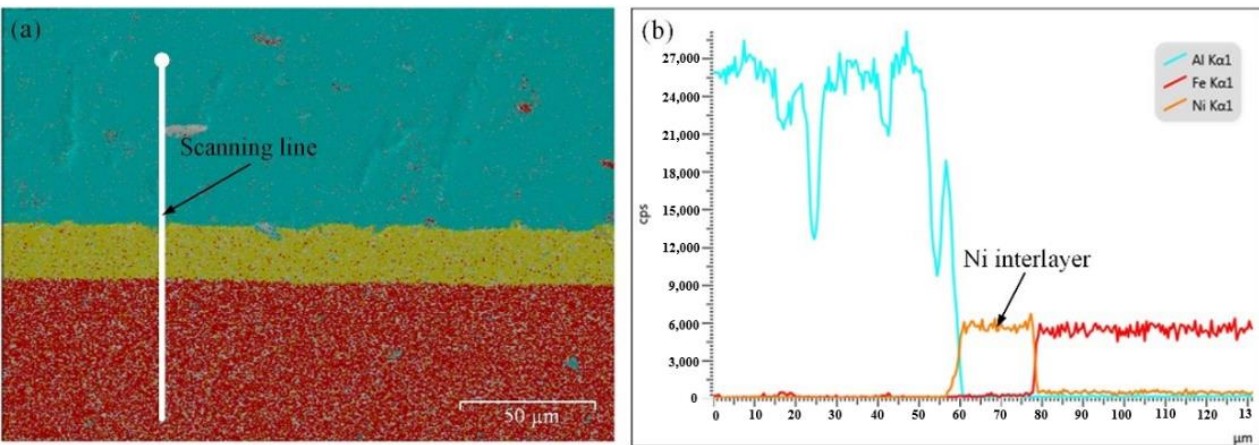

**Figure 6.** EDS analysis of region C at 800 rpm: (**a**) area scan and (**b**) line scan.

In this study, the spectrum obtained from the XRD analysis was compared with the standard spectrum, and then, the phase composition of the joint was determined through Jade 6.5 software (MDI Jade 6.5, Materials Data, California, United States), manufacturer name, city, country). As shown in Figure 7, Al, γ-Fe, Ni and AlNi₃ were detected by the XRD analysis near the lap interfaces of the joints at 800 and 1200 rpm. The Ni atoms can be infinitely solid and dissolved into the steel according to the study by Kundu et al. [30], but there is still residual Ni material at the lap interface. Thus, the Ni element is scanned, as displayed in Figure 7. According to the EDS analysis, the Ni interlayer effectively inhibits the diffusion of Al and Fe atoms; thus, the formation of Al-Fe IMCs near the interface area fails. According to the Al-Ni phase diagram, it is known that the Al-Ni IMCs can be produced in the solid-state welding process [21]. In this study, the Ni element is diffused to the aluminum side (Figures 3 and 4), resulting in a metallurgical reaction between Al and Ni atoms near the lap interface. Therefore, the AlNi₃ IMCs rather than the Al-Fe IMCs is formed at the lap interface and is then scanned by the XRD method.

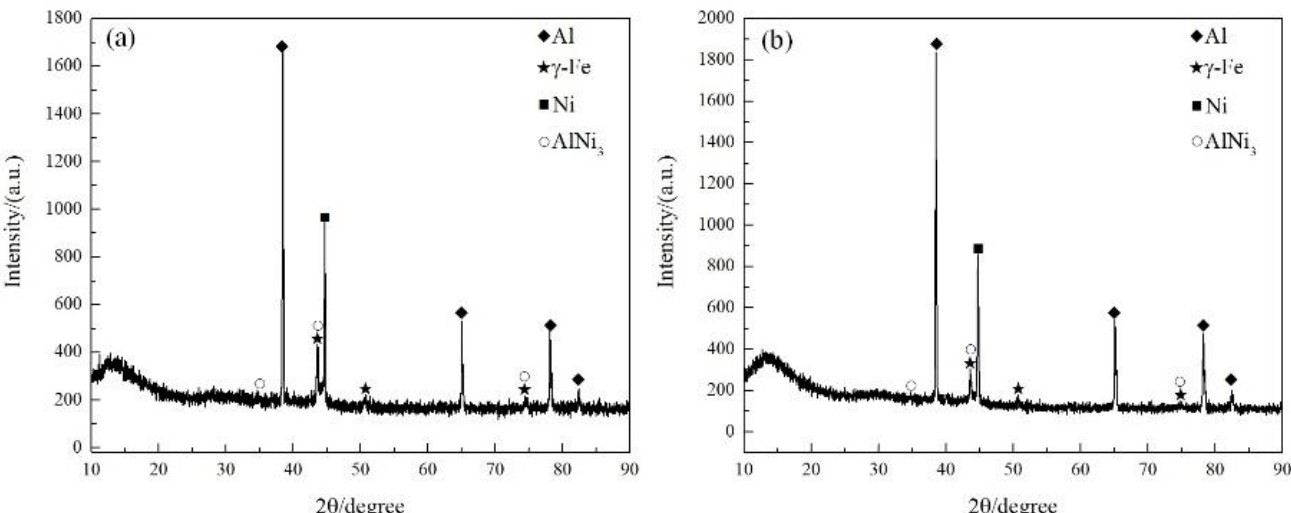

**Figure 7.** XRD analysis of the lap interface: (**a**) 800 rpm and (**b**) 1200 rpm.

*3.3. Microhardness*

In this study, the microhardness of the joint was analyzed, and the measured positions are shown in Figure 3a. In the horizontal direction, the microhardness distributions in the middle of the upper aluminum alloy sheet under different rotating speeds are shown in

Figure 8a. The microhardness of the 2024 aluminum alloy is mainly affected by grain size, precipitate phase and dislocation density [31], and the heated zones including heat-affected zone (HAZ), thermo-mechanically affected zone (TMAZ) and stir zone (SZ) always have lower microhardness compared with BM. According to the Hall–Petch relationship [20], the material microhardness increases with decreasing grain size. The grain size in SZ can be greatly refined due to the dynamic recrystallization, and the grain size in HAZ can be coarsened under the action of the thermal cycle; thus, the microhardness of SZ is higher than that of HAZ, resulting in the "W"-shaped distribution as displayed in Figure 8a. Under the rotating speeds of 800, 1000 and 1200 rpm, the maximum microhardness values of SZ are 129.6, 135.9 and 137.3 HV, respectively. This phenomenon shows that the grain size in SZ decreases when the rotating speed varies from 800 to 1200 rpm.

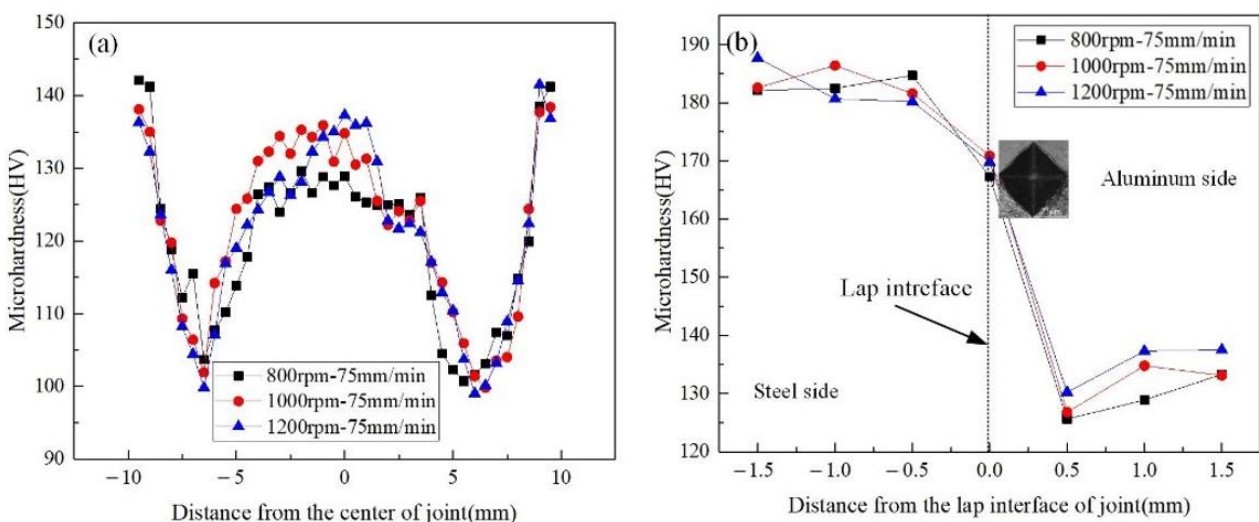

**Figure 8.** Microhardness distributions of joints: (**a**) horizontal direction and (**b**) thickness direction.

Figure 8b shows the microhardness distributions along the joint thickness direction under different rotating speeds. The "Z"-shaped distribution is observed in Figure 8b, because the microhardness of the 304 stainless steel BM is higher than that of the 2024 aluminum alloy BM, and the microhardness of the 2024 aluminum alloy is decreased under the heating effect of the stirring tool. According to the results of Xiong et al. [32], it is known that the microhardness of $AlNi_3$ is smaller than that of the 304 stainless steel BM. Moreover, during the microhardness measurement, the diagonal length of the indention (Figure 8b) is much larger than the 2 μm thickness of the $AlNi_3$ IMCs layer; thus, the measured point at the lap interface covers the two BMs and the IMCs layer. Based on these above two reasons, the microhardness of the point at the lap interface is higher than that of the upper sheet and smaller than that of the lower sheet. The microhardness of the point at the interface at 1200 rpm is slightly higher than that at 800 rpm, because the IMCs layer thickness at 800 rpm is smaller than that at 1200 rpm. At the aluminum side, it is seen that the microhardness values are changed when the rotating speed varies, which is due to the differences in grain size, precipitate phase and dislocation density of the material [31].

*3.4. Tensile Shear Strength*

In this study, the tensile shear load of the Al/steel FSLW joint was tested, and the values are presented in Figure 9. In order to show the advantages of this study, the tensile shear loads of the FSLW joints of the dissimilar aluminum and steel materials reported in some studies [4,20,33–35] were compared. As mentioned above, controlling the heat input and changing the IMCs type are the two important ways to enhance the tensile strength of the dissimilar aluminum and steel alloys FSLW joint. Dong et al. [20] stated that the tensile load of the Al/steel joint with the Zn interlayer was higher than that without the Zn

interlayer, and the maximum value was 2.5 kN/mm. Compared with the tensile load in this study, 2.5 kN/mm is much smaller, which is because the tool pin does not insert into the lower steel sheet. Wan et al. [22] and Geng et al. [28] fabricated an Al/steel FSLW joint under a 0.2 mm plunging depth of the tool pin into the lower steel sheet, and their obtained tensile loads of the joints were respectively 3.33 and 3 kN/mm. According to the literature reported by Xiong et al. [36], it is known that controlling the plunging depth of the tool pin into the steel is considered as a necessary prerequisite for producing a sound joint, and the suitable plunging depth should be no more than 0.1 mm for the Al/steel FSLW joint. This may be the reason that the tensile loads of the joints obtained by Wan et al. [22] and Geng et al. [28] are both lower than the maximum load in this study. Thus, there are two ways that need to be considered together regarding the viewpoint of obtaining a high strength of the Al/steel FSLW joint. One is to using a slight plunging depth of the tool pin into the lower steel sheet for controlling the heat input, the other is adding the interlayer for changing the IMCs type instead of the Al-Fe IMCs. It is known from Figure 9 that the relatively high tensile shear load of the joint is obtained in this study due to the avoidance of Al-Fe IMCs and the rational Al-Ni IMCs layer thickness, which proves that the 0.3 mm plunging depth of the tool pin into the lower steel sheet and the addition of 0.2 mm thick Ni interlayer are both positive on heightening the tensile strength of the Al/steel FSLW joint. Moreover, when the rotating speed of the stirring tool increases from 800 to 1200 rpm, the tensile shear load of the Al/steel FSLW joint decreases. These load values of the joints at 800, 1000 and 1200 rpm are respectively 3.95, 4.20 and 4.34 kN/mm. In this study, the tensile shear strength was calculated by dividing the tensile shear load by the specimen width. For the specimen with a 20 mm width, the maximum tensile strength of the joint is 217 MPa, and the value is 47.2 percent of that of the 2024 aluminum alloy BM.

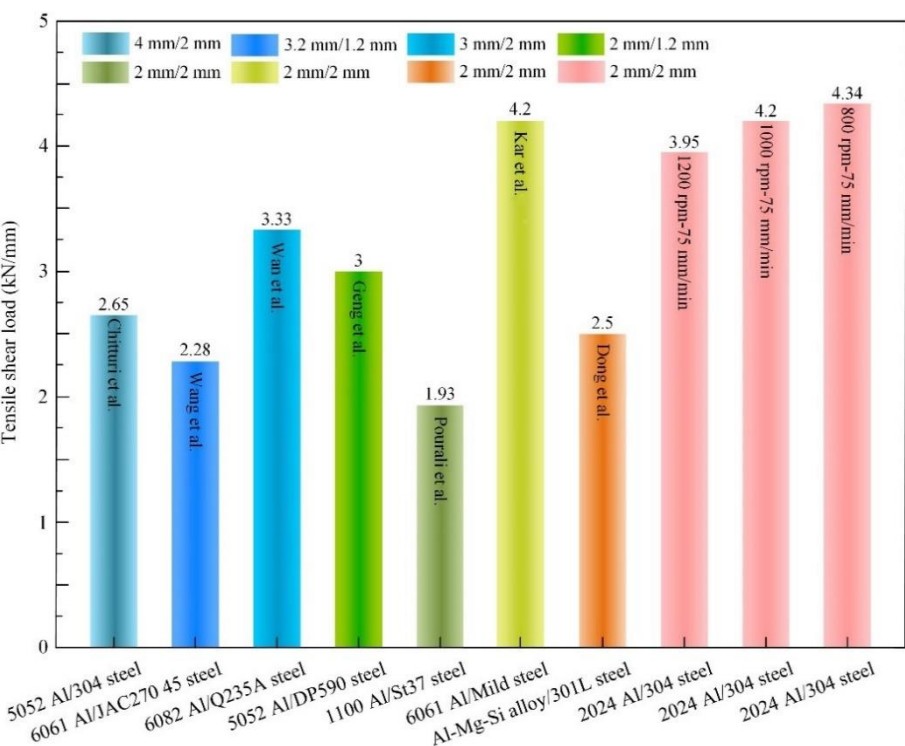

**Figure 9.** Tensile shear loads of joints of this study and reported literature. Data from [4,20,22,28,29,33,35].

In this study, after exerting the tensile load, cracks initiate at the lap interface and propagate along the interface, finally causing shear fracture of the joint. The shear fracture mode means that the characteristics of the lap interface greatly affect the tensile shear properties of the joint. As mentioned above, the hook structure (Figure 2) and the mechanical occlusion (Figures 3 and 4) are formed at the lap interface, which greatly influences the

joint strength. The hook structure at the lap interface of the dissimilar materials FSLW joint may or may not be positive to the joint strength [37,38] because this hook structure has two features of hindering crack propagation and inducing stress concentration. Hindering crack propagation is beneficial to enhancing the joint strength, but inducing stress concentration is detrimental to the joint strength. In this study, the joint at 800 rpm has no hook structure and has a higher strength compared with the other two joints with the hook structure, which shows that inducing the stress concentration plays a dominant role from the viewpoint of heightening the joint strength. The mechanical occlusion at the lap interface plays a role similar to the hook structure, where the IMCs layer thickness is bigger than the other regions at the lap interface (Figure 5). The increased IMCs layer thickness is detrimental to hindering crack initiation and propagation. From Figures 2–4, it is known that with increasing the rotating speed from 1000 to 1200 rpm, the hook structure, the mechanical occlusion and the IMCs layer thickness at the lap interface are all enhanced, thereby resulting in a further decrease in the joint strength. In other words, when the rotating speed increases from 800 to 1200 rpm, the tensile strength of the joint decreases, and a maximum value of 4.34 kN/mm is obtained at 800 rpm.

Choosing the steel side of the joint at 800 rpm after the test as the research object, the fracture morphology was observed, and the XRD analysis was performed, as displayed in Figure 10. The arc trace is observed on the fracture surface in Figure 10a and agrees with the mechanical occlusion at the lap interface (Figures 3 and 4), which results from the tool pin constantly scraping the steel interface. For the FSLW process, the welding temperature is the key factor on influencing the interfacial characteristics and then the joint strength. However, the welding temperature, which is decided by many factors such as the heat transfer area and the plunging depth of the tool pin, is impossible to keep as constant in the whole welding process. This is because the heat transfer area varies along the welding direction, and the plunging depth of the tool pin is difficult to keep constant because the thickness of the plate to be welded fluctuates in a small range. Moreover, the plunging depth of the tool pin varying in a small range influences the stirring region at the lap interface. Then, the slightly changed temperature and stirring region at the interface influence the interfacial characteristics of the joint. In this study, the SEM image in Figure 10a was taken from the tensile specimen, whose location was adjacent to the keyhole near the ending point of weld. At this location, the temperature varied more violently compared with the middle part of the weld according to the reported results by Ji et al. [39], which was due to the greatly reduced heat transfer area. Thus, the width of the weld at the interface varies from bottom to up in the image in Figure 10a. Moreover, some other materials are tightly bonded on the steel surface (Figure 10a), and these materials consist of aluminum and nickel alloys rather than Al-Ni IMCs according to the XRD results in Figure 10b. This phenomenon reveals that the bonding strength between the upper aluminum alloy and the Al-Ni IMCs layer is larger than that between the steel and the Al-Ni IMCs layer. On the fractured surface at the steel side, two typical regions were chosen and enlarged to further analyze the fracture morphology by SEM, and the results are shown in Figure 10c,d. Cleavage steps and dimples are both observed; thus, the joint is characterized by the brittle–ductile mixed fracture. This brittle–ductile mixed fracture of the Al/steel FSLW joint is also reported by Chitturi et al. [25].

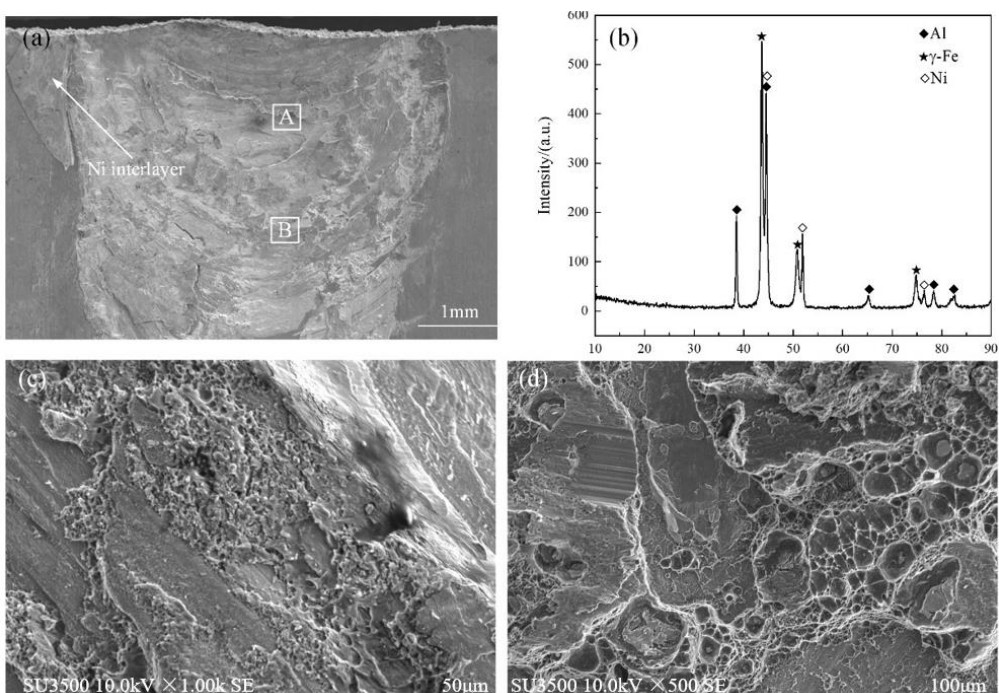

**Figure 10.** Characteristics of the fracture surface at the steel side of the joint at 800 rpm: (**a**) fracture morphology and (**b**) XRD analysis; (**c**,**d**) enlarged views of regions A and B marked in Figure 10a.

## 4. Conclusions

Under the addition of Ni foil as an interlayer and the very slight plunging depth of the tool pin into the lower steel sheet, 2024 aluminum alloy and 304 stainless steel were successfully joined by FSLW. The relationship between the lap interface characteristics and the joint mechanical properties was discussed. The following conclusions were extracted.

(1) Due to the tool pin plunging into the lower steel sheet, the hook structure and mechanical occlusion were formed at the lap interface. With increasing the rotating speed of the stirring tool, the height of the hook structure was increased, and the mechanical occlusion was enhanced due to the elevated welding temperature and improved material flow behavior.

(2) The Ni interlayer at the lap interface changed the IMCs type from Al-Fe IMCs to Al-Ni IMCs. The obtained AlNi$_3$ IMCs layer thickness at the lap interface was smaller than 2 μm, which was beneficial to heightening the strength of the Al/steel FSLW joint.

(3) Under the constant welding speed of 75 mm/min, the tensile shear load of the joint decreased when the rotating speed increased from 800 to 1200 rpm, and the maximum value of 4.34 kN/mm was obtained at 800 rpm. The welding joint shear fractured along the lap interface and presented a brittle–ductile mixed fracture.

**Author Contributions:** Conceptualization, P.G., Y.Z. and Y.Y.; literature search, R.G. and Z.Y.; data collection, R.G. and Z.Y.; figures, R.G. and Z.Y.; data analysis, P.G., J.L., Y.Y. and R.G.; methodology, R.G. and J.L.; writing—original draft preparation, R.G. and J.L. All authors have read and agreed to the published version of the manuscript.

**Funding:** This project was supported by the National Natural Science Foundation of China (52174366) and the Education Department Foundation of Liaoning Province (no. JYT2020051).

**Institutional Review Board Statement:** Not applicable.

**Informed Consent Statement:** Not applicable.

**Data Availability Statement:** Not applicable.

**Conflicts of Interest:** The authors declare no conflict of interest.

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
