# Peer review of "Interface Characteristics and Mechanical Properties of 2024 Aluminum Alloy and 304 Stainless Steel Dissimilar Alloys FSLW Joint with Ni Interlayer"

_metals, doi:10.3390/met12101574_

Round 1

Reviewer 1 Report

1.     Is the thickness of the Ni interlayer uniform along the interface, and does it has any effect on the strength of the joints?

2.     The EDS line scan revealed a certain amount of intermixing zone, but in the SEM or EDS micrographs, it doesn’t identify or justify.

3.     What is the temperature difference between 800 rpm and 1200 rpm welds, and in what way authors were tried to find the different intermetallic compounds in fig 7.

4.     In fig 10a, the width of the nugget seems to be varied from bottom to up in the image, is the entire weld length of the nugget the same or varies? Please provide the evidence with suitable reasons.

5.     Hardness of the three different speeds is quite different due to strain hardening or the Ni interlayer effect.

6.     In fig 1, the Ni interlayer is not clear; it is suggested to make it clear.

7.     Is the tilting angle required for Ni interlayer welding or is it possible to weld without tilting?

8.     Many studies were already reported that there were no serious IMCs in the Al/Fe welds by solid-state welds.. Why interlayer welds need to be studied.

9.     The scientific discussion of the manuscript should be improved.   

Reviewer 2 Report

This is an excellent fit for Metals. Here some comments to further improve the quality of this work:

-check language and for typos

-interlayer in form of a foil? I would mention this in the abstract

-4.34 kN is what percentage of the base materials? I would also mention this in the abstract

-line 40-41: I would also mention diffusion bonding for which interlayers are actually pretty common for stainless steel dissimilar bonds (https://doi.org/10.3390/met10050613)

-The discussion/state of the art is a bit short and there is no direct comparison between past studies. Maybe summaries these past studies in a table and compare them by the tensile test value, etc. that was obtained…-excellent you did this in Fig. 9 – good work, I suggest using reference author names in Fig. 9 as well for better orientiation…

-the stirring tool was made of which material?

-Experimental description is fine with me

Round 2

Reviewer 1 Report

The quality of the revised version is improved.

Reviewer 2 Report

Please double check that you send the right revised manuscript.

I did not see Point 4 addressed. The rest seems fine with me.
